**Data Availability Statement:** Data are available on Figshare at doi.org/10.6084/m9.figshare. 12691856.

# Working nights and lower leisure-time physical activity associate with chronic pain in Southern African long-distance truck drivers: A cross-sectional study

**Antonia L. Wadley**[1]*, **Stella Iacovides**[1], **Johanna Roche**[1], **Karine Scheuermaier**[1], **W. D. Francois Venter**[2], **Alinda G. Vos**[2,3], **Samanta T. Lalla-Edward**[2]

**1** Brain Function Research Group, School of Physiology, Faculty of Health Sciences, University of the Witwatersrand, Johannesburg, South Africa, **2** Ezintsha, A Sub-Division of Wits Reproductive Health and HIV Institute, Faculty of Health Sciences, University of the Witwatersrand, Johannesburg, South Africa, **3** Julius Center for Health Sciences and Primary Care, University Medical Center Utrecht, Utrecht, The Netherlands

\* antonia.wadley@wits.ac.za

## Abstract

### Background

In South Africa, the trucking industry employs over 70,000 people and the prevalence of chronic pain in this occupational group was reported at 10%. We investigated factors associated with chronic pain in truck drivers including mental health, physical activity, and sleep, as no study has done so.

### Methods

Southern African male, long-distance truck drivers were recruited at truck stops in Gauteng and Free State Provinces, South Africa (n = 614). Chronic pain was defined as pain present for at least the last three months. Depressive symptoms were assessed with the Patient Health Questionnaire-9, post-traumatic stress disorder with the Posttraumatic Stress Disorder Checklist for DSM-5 (PCL-5), exposure to traumatic events with the Life Events Checklist-5 (LEC-5) and daytime sleepiness with the Epworth Sleepiness Scale. Sleep quality was measured on a four-point Likert scale. Leisure-time physical activity was measured using the Godin-Shephard leisure-time physical activity questionnaire. Associations between these factors, demographic factors and chronic pain were investigated.

### Results

Multivariate analysis showed that working ≥ 2 nights/week (OR = 2.68, 95% CI = 1.55–4.68) was associated with chronic pain and physical activity was protective (OR = 0.97, 95% CI 0.95–0.98). In an exploratory analysis, greater depressive symptoms (p = 0.004), daytime sleepiness (p = 0.01) and worse sleep quality (p = 0.001) was associated with working ≥ 2 nights/week. Lower leisure-time physical activity was associated with worse sleep quality (p = 0.006), but not daytime sleepiness or depressive symptoms (p>0.05).

**Funding:** This work was funded by North Star Alliance through a research and implementation grant received from the Ministry of Foreign Affairs of The Netherlands managed by the Royal Dutch Embassy of Mozambique. The Amsterdam Institute for Global Health and Development and Wits RHI held separate contracts with North Star Alliance (AIGHD's grant reference: 0068 North Star – NSCDP; RHI's grant number: D1404070). The views of this study are those of the authors and do not necessarily reflect the views of any of the funders or the South African and Dutch governments.The funders had no role in study design, data collection and analysis, decision to publish, or preparation of the manuscript.

**Competing interests:** The authors have declared that no competing interests exist.

## Conclusions

There is a clear relationship between working nights and activity levels, and chronic pain, sleep quality, and depression in truck drivers.

## Introduction

Trucking is a major industry and employer worldwide. In the US alone the trucking industry generated almost $800 billion in revenue in 2018 and employed 3.5 million drivers [1]. In South Africa, due to the lack of waterways and the poor rail services, road transport is essential to the national economy [2], and transport workers represent a large group of employees: over 70,000 truck drivers, who are mostly men [3].

As truck drivers contribute significantly to economies, both worldwide and in South Africa, ensuring the health of this occupational group is important [4]. This task requires effort, though, as truck drivers are at a greater risk of a multitude of physical and mental health disorders compared to the general population [5–7]. Risk factors for non-communicable diseases such as cardiovascular disease, diabetes and poor mental health easily result from long-distance driving due to lack of physical activity, inability to eat healthily, and reduced or disrupted sleep [8,9]. Obesity and reduced physical activity are also risk factors for chronic pain [10,11], which is one of the most common causes of disability and reduced quality of life [12,13]. Studies of truck drivers' health to date have tended to focus on cardiovascular health [14,15] yet chronic pain affects more Americans than cardiovascular disease, diabetes and cancer combined [16].

The prevalence of pain in truck drivers ranges from 27–81% [8,17–20] and long-distance truck drivers have a higher prevalence of pain than short-haul drivers [19]. In addition to road accident-related injuries, prolonged sitting and exposure to vibration, truck drivers are at high risk of non-traffic-related incidents too as work tasks involve lifting and unloading of cargo [18–20]. The development of chronic pain is more complicated than risk of injury, however, and is strongly associated with psychosocial factors such as depression and previous trauma [21]. Of the studies that have looked at psychosocial factors on drivers' pain, the factors have tended to be work-related e.g. job satisfaction, perceived work stress, effort-reward balance [19,20]. Even when social support has been included it has been in terms of social support from employers and colleagues rather than from individuals' personal lives [19]. In essence, studies investigating pain in truck drivers have not explored psychosocial factors from an holistic point of view.

In addition to assessing trait (rather than just work-related) psychosocial factors, a comprehensive investigation of factors associating with pain in truck drivers would also include sleep and physical activity. Truck drivers work night shifts, likely reducing sleep and causing irregular sleep/wake times, and have low physical activity due to long hours working [9]. Pain and sleep have a reciprocal relationship [22,23]. Together, depressed mood, disrupted sleep and reduced physical activity often associate with pain and reduced quality of life in general populations [24,25]. The associations of these factors in a cohort of truck drivers has not been examined, however.

Healthcare programmes specifically designed for truck drivers in sub-Saharan Africa have been operating for the last 30 years [26]. These programmes have typically focused on HIV testing and counselling and more recently have offered primary health care services too [26]. Roadside Wellness Centres (RWC) and mobile HIV clinics have been set up along driving routes in South Africa to provide such services. An analysis of the up-take of RWC services

recommended that the services provided at the RWC needed to be designed for the needs of truck drivers [27]. In order to design services for long-distance drivers that promote health and quality of life, factors that associate with chronic pain need to be determined. As such, the aim of this study was to assess associations with a variety of factors including psychosocial, sleep, physical activity and driving-related factors with the prevalence of chronic pain in truck drivers. As long-distance drivers are more likely to work nights than short-distance drivers, and we wished to investigate the association between sleep disruption and pain, we recruited long-distance drivers.

## Materials and methods

In this cross-sectional analysis of 614 male Southern African truck drivers, we investigated factors associated with chronic pain. The study was designed to determine the general health of male truck drivers in South Africa and comprehensive descriptive analyses have been published elsewhere [28]. In brief, the cross-sectional study was conducted during October 2016 and March 2017. Data were collected at North Star Alliance (NSA) wellness clinics or the nearby truck stops in Pomona in the Gauteng province, and Bloemfontein, which is in the Free State province, of South Africa. These stops were chosen because of the close proximity of the wellness clinic for referral and the high volume of truck drivers in the vicinity. Recruitment also took place in Soweto, Gauteng Province which was a busy truck stop but not located near the NSA wellness clinic, in order to include more South Africans in the survey. Recruitment took place from outside the clinic or at the truck stops on weekdays during clinic operating times, which varied from 4pm (Bloemfontein) to 8pm (Pomona). Recruitment in Soweto took place during weekdays and also on weekends until 3pm. There were several methods used to recruit truck drivers: invitation cards were handed out at the truck stops to individuals and groups, cards were placed on windscreens and at recruiting companies. Sometimes a group would form when a recruiter was speaking to one or two drivers. As such, it was not possible to record accurately the number of drivers agreeing to participate and those declining. Inclusion criteria were males of 18 years or older, full-time, long-distance truck drivers and able to provide informed consent. Participants were given 150 ZAR and a T-shirt for taking part.

Ethics clearance was obtained from the University of the Witwatersrand Human Research Ethics Committee (Medical), which adheres to the principles of the Declaration of Helsinki [clearance number M160760] and all participants gave written informed consent.

Trained nurses measured height, weight and tested for HIV using standard Department of Health procedures. Non-fasting blood was drawn to measure serum C-reactive protein.

### Measures

**Chronic pain.**   We used a simple but standard method for determining chronic pain: chronic pain was classified as having had pain at any location for at least the last three months [29]. Participants were only asked one question: they were asked to indicate any locations of pain from a list of body parts, where they had had pain or discomfort for more than three months. The data on location of pain sites has been presented elsewhere [28]. Here the binary data of having chronic pain or not were used.

**Physical activity.**   Physical activity was measured using the Godin-Shephard leisure-time physical activity questionnaire (GSLTPAQ) [30]. Number of 15-minute episodes of strenuous, moderate and light exercise per week were recorded. A leisure-time physical activity score was calculated by multiplying the strenuous episodes by nine, the moderate episodes by five and the light episodes by three. The questionnaire has been previously used in African contexts [31,32].

**Sleep and daytime sleepiness.**   Subjective sleep parameters were assessed using the Pittsburgh Sleep Quality Index (PSQI) [33], which has been previously validated in the South African population [34]. The PSQI is a self-administered questionnaire composed of 19 items, which measure multiple dimensions of sleep during the previous month. A PSQI global score is calculated from the questionnaire and ranges from 0–21, with higher scores indicating poorer sleep quality. The global score is comprised of seven component scores (subjective sleep quality, sleep latency, sleep duration, habitual sleep efficiency, sleep disturbances, use of sleeping medication, and daytime dysfunction) which can be analyzed separately [35]. For the purpose of this study, we examined the overall sleep quality as assessed by the first component of the PSQI, which refers to question 9, where the participants are asked to rate their subjective sleep quality overall during the past month as "very good", "fairly good", "fairly bad" or "very bad" [35].

Excessive daytime sleepiness was measured using the Epworth Sleepiness Scale (ESS) [36]. This is a self-administered and validated scale that aims to assess the presence and degree of daytime sleepiness, which has been previously validated in the South African population [34]. The respondent is asked to answer how likely they are to dose of or fall asleep in eight different scenarios on a 4-point Likert scale of 0 ("would never dose") to 3 ("high chance of dozing"). Possible scores range from 0 to 24 and in the general population, a score > 10 means excessive daytime sleepiness [36,37].

**Mental health.**   Depressive symptoms were measured using the widely-used Patient Health Questionnaire (PHQ-9) [38]. This is a 9-item questionnaire with answers ranging from 0 (Not at all) to 3 (Nearly every day). The answers are summed resulting in a score ranging from 0–27. Cut-off points for symptom severity are 5 for mild, 10 for moderate, 15 for moderately severe, and 20 for severe depression. The tool has been successfully validated in primary health care populations in South Africa [39,40].

Post-traumatic stress disorder (PTSD) was assessed using the Posttraumatic Stress Disorder Checklist for DSM-5 (PCL-5) [41]. This is a 20-item scale where participants answer on a 5-point Likert scale from 0 (Not at all) to 4 (Extremely) to a range of DSM criteria for PTSD, including presence of intrusive symptoms, negative changes in cognitions and mood and alterations in arousal and reactivity [42]. A total symptom severity score is calculated by summing the item scores resulting in a score ranging from 0–80. Cut-off scores for probable PTSD can range between 23–38 depending on the population the scale was validated in [43–46]. The cut-off was 33 in an African population from Zimbabwe [47], the country from which the majority of the truck drivers came, so we used this cut-off score.

The life events checklist (LEC-5) [48] is a tool often used alongside the PCL-5, and determines exposure to traumatic life events. There are 16 stressful life events such as 'Fire or explosion', "Sexual assault" and "Sudden unexpected death of someone close to you" with an additional item for "Any other very stressful event or experience". Respondents answer with 'Happened to me', 'Witnessed it', 'Learned about it', 'Not sure', 'Doesn't apply'. For the purposes of this study, we determined that the total number of events with the response recorded as 'happened to me' as direct exposure, and we determined the number of events with the response recorded as 'witnessed it' as indirect exposure. We report both but used the former in our analyses.

## Driving-related factors

Factors related to participants' driving were also recorded. These factors included number of years driving, average number of driving days per month, average number of hours driving per day. The number of working nights per week was determined by asking drivers how many

times a week they drove for at least three hours between 10pm and 6am. Stressful events whilst working were determined by asking drivers if they had had an accident, and if so, the number of accidents (of any severity) they had had. Drivers were also asked if they carried dangerous cargo, such as explosives, flammable gases, combustible solids or radioactive material.

### Data analysis

The analysis was run on GraphPad Prism (version 8.0.0, San Diego, California USA). Data were checked for normality and were non-parametric. As such, data are presented as median (interquartile range, IQR) and non-parametric statistical tests were run. The presence of chronic pain was the dependent variable. In univariate analysis, a Fisher's exact test was used to analyze 2x2 tables for its exact p value but Chi-square tests were used for categorical variables with more groupings. Variables routinely associated with pain, that is age, body mass index (BMI) and depressive symptoms [49], or variables achieving a p value <0.1 on univariate analysis were added to the multivariate logistic regression. In addition, as there is a known relationship between sleep, pain and depression [25], we ran exploratory analyses including sleep quality and depression levels with the significant main effects found in our model (working nights and leisure-time physical activity). That is, to investigate the hypothesis of a relationship between pain, sleep, depression and activity, we investigated whether 1) sleep quality, sleepiness and depression levels were actually worse in those working two or more nights a week compared to those working one or fewer nights a week and 2) whether depression and sleep were worse in those with lower leisure-time physical activity. P values <0.05 were considered to be significant.

## Results

Data from six hundred and fourteen male Southern African drivers were analysed. A description of the entire cohort has already been presented [28]. In brief, the median age of the male truck drivers was 37 years (IQR 31–42), they had mostly completed secondary education (89%, 486/614) and two-thirds classified as overweight or obese (BMI ≥ 25, 65%, n = 417/614). The majority of the drivers were Zimbabwean (63%, 384/614), with the remainder being mostly South African (20%, 123/614) or Zambian (7%, 45/614). Of the 614, 63 (10%) had chronic pain. Characteristics of those with and without chronic pain are shown in Table 1.

Depressive symptoms (mild or greater, ≥5 on PHQ-9) occurred in 43% (27/63) of participants with chronic pain, and 32% (178/548) without chronic pain (Chi-squared test for trend, p = 0.11). PTSD was present in 6% (4/63) of those with chronic pain and 4% (20/549) of those without. There was no difference between the groups (Fisher's exact, p = 0.3). At least one traumatic life event had been directly experienced ("happened to me") by 75% (456/612) of the drivers. When witnessed traumatic life events were added to those directly experienced ("happened to me" and "witnessed it"), 95% (582/612) of the drivers reported previous exposure.

### Factors associating with chronic pain

Table 1 also shows results of the univariate analysis to determine associations with chronic pain. Age, BMI, and HIV status were not associated with having pain (p>0.05). Lower leisure-time physical activity levels and working nights were associated with having chronic pain (p<0.05). Smoking, and traumatic life experiences as measured by the LEC-5 achieved p values <0.1 and so were included in the multivariate model.

Table 2 shows the results of the multivariate model. Leisure-time physical activity levels and working nights remained significant (p<0.001), whereby higher levels of activity were associated with lower likelihood of chronic pain (OR = 0.97, 95% CI 0.95–0.98) and working ≥2

**Table 1. Characteristics and comparison of the truck drivers with and without pain.**

| | Chronic pain | No chronic pain | P value |
|---|---|---|---|
| | n = 63 | n = 551 | |
| Age (years) * | 37.0 (31.0–41.5) | 37.0 (31.0–42.0) | 0.71[†] |
| BMI*[$] | 27.5 (24.5–31.7) | 27.1 (24.1–30.5) | 0.23[†] |
| Level of education, n (%)[$] | | | 0.80[‡] |
|   Primary or less | 4 (6) | 47 (9) | |
|   Some secondary | 55 (87) | 431 (78) | |
|   Tertiary | 4 (6) | 50 (9) | |
| Smoking, n (%)[$] | 4 (6) | 86 (16) | 0.06[α] |
| Activity score* | 0 (0–27) | 17 (0–27) | 0.0001[†] |
| HIV-positive, n (%)[$] | 7 (11) | 45 (8) | 0.48 [α] |
| C-reactive protein (mg/l)* [$] | 1.30 (0.50–3.10) | 1.30 (0.50–2.75) | 0.58[†] |
| PHQ-9, n (%)[$] | | | 0.11[‡] |
|   Minimal | 36 (57) | 370 (68) | |
|   Mild | 20 (32) | 135 (25) | |
|   Moderate or greater | 7 (11) | 43 (8) | |
| PCL-5*[$] | 6 (2–14) | 0 (5–13) | 0.49[†] |
| LEC-5 *[$] | 3 (1–4) | 2 (0–3) | 0.07[†] |
| **Sleep-related** | | | |
| Sleep quality, n (%)[$] | | | 0.94[‡] |
|   Very good | 29 (46) | 217 (40) | |
|   Fairly good | 20 (32) | 222 (40) | |
|   Fairly bad | 10 (16) | 90 (16) | |
|   Very bad | 4 (6) | 18 (3) | |
| ESS score* | 6 (4–8) | 5 (2–9) | 0.30[†] |
| **Driving-related** | | | |
| Number of years driving* | 11 (4–15) | 8 (5–14) | 0.49[†] |
| Driving days per month, n (%) | 20 (20–24) | 20 (15–24) | 0.37[†] |
| Hours driving per day*[$] | 10 (9–12) | 10 (8–12) | 0.44[†] |
| Working nights per week, n (%)[$] | | | 0.0002 [α] |
|   0–1 | 27 (44) | 371 (68) | |
|   2 or more | 35 (56) | 174 (32) | |
| Accident whilst driving, n (%)[$] | 14 (22) | 102 (19) | 0.73 [α] |
| Number of accidents whilst driving*[$] | 0 (0–0) | 0 (0–0) | 0.60[†] |
| Carry dangerous cargo, n (%) | 17 (27) | 200 (36) | 0.21 [α] |

Median (IQR)

[†] Mann Whitney

[‡]Chi-square test for trend

[α] Fisher's exact.

PHQ-9 = Patient Health Questionnaire-9; PCL-5 = Posttraumatic Stress Disorder Checklist for DSM-5; LEC-5 = Life Events Checklist-5; ESS = Epworth Sleepiness Scale.

PHQ-9 categories: minimal = score of 0–4, mild 5–9, moderate or more $\geq$ 10.

Rounding resulted in some percentages not adding to 100%.

[$]Missing data. BMI: chronic pain = 1, no chronic pain = 9; Education: no chronic pain = 23; Smoking: without chronic pain = 3; HIV: no chronic pain = 12; CRP: chronic pain = 2, no chronic pain = 23; PHQ-9: no chronic pain = 3; PCL-5: no chronic pain = 3; LEC-5: no chronic pain = 2; Sleep quality: no chronic pain = 4; Hours driving per day: no chronic pain = 7; Working nights per week: chronic pain = 1, no chronic pain = 6; Accidents whilst driving: chronic pain = 1; no chronic pain = 7; Number of accidents whilst driving: chronic pain = 1, no chronic pain = 9

**Table 2. Results of multivariate logistic regression model for the presence of pain.**

|  | *Odds ratio* | *95% confidence intervals* | *p-value* |
|---|---|---|---|
| Age (years) | 0.98 | 0.95–1.02 | 0.30 |
| BMI | 1.02 | 0.97–1.08 | 0.43 |
| Smoking | 0.44 | 0.13–1.20 | 0.13 |
| Activity score | 0.97 | 0.95–0.98 | 0.0008 |
| PHQ-9 | 1.15 | 0.76–1.70 | 0.51 |
| LEC-5 | 1.05 | 0.94–1.17 | 0.36 |
| Working ≥2 nights per week | 2.68 | 1.55–4.68 | 0.0004 |

BMI = Body mass index; PHQ-9 = Patient Health Questionnaire-9; LEC-5 = Life Events Checklist-5.

nights/week was associated with higher likelihood of chronic pain (OR = 2.68, 95% CI = 1.55–4.68).

## Exploratory analysis

As pain, sleep and depression are tightly interrelated [25], we further hypothesized that the relationship between working nights and having chronic pain may be through worse sleep quality and higher depressive symptoms. Having found that working nights was associated with higher odds of having chronic pain, we verified whether sleep was worse in those working nights. Table 3 shows that sleep quality and daytime sleepiness were indeed worse in those working two or more nights a week (respectively, p<0.001 and p<0.01). We also found higher prevalence of mild/moderate depressive symptoms in those working ≥2 nights/week compared to those working <2 (40% vs 30%, p = 0.004, see Table 3).

Secondly, as sleep, depression and physical activity are also interrelated and share common biochemical and neural pathways [50], we hypothesized that the relationship between lower

**Table 3. Comparison of depression and sleep-related measures between those working zero or one night a week to those working two or more nights per week.**

|  | Working 0 or 1 night/week | Working ≥2 nights/week | P value |
|---|---|---|---|
|  | n = 398 | n = 209 |  |
| **Mental health-related** |  |  |  |
| PHQ-9[$] |  |  |  |
| Minimal | 277 (70) | 123 (60) | 0.004[‡] |
| Mild | 96 (24) | 58 (28) |  |
| Moderate | 25 (6) | 25 (12) |  |
| **Sleep-related** |  |  |  |
| Sleep quality[$] |  |  |  |
| Very good | 176 (44) | 67 (32) | 0.001[‡] |
| Fairly good | 153 (39) | 87 (42) |  |
| Fairly bad | 56 (14) | 42 (20) |  |
| Very bad | 11 (3) | 11 (5) |  |
| ESS score | 5 (2–9) | 6 (3–10) | 0.0096[†] |

Data presented as n (%) or median (IQR).

[†] Mann Whitney; [‡] Chi-square test for trend.

[$] Number of missing data. PHQ-9 ≥2 nights n = 3; Sleep quality: 0-1nights n = 2, ≥2 nights = 2

PHQ-9 = Patient Health Questionnaire-9; ESS = Epworth Sleepiness Scale.

activity and chronic pain may also be through worse sleep quality and higher depressive symptoms. Indeed, sleep quality was worse in those who had lower activity levels (Kruskal Wallis, KW statistic = 12.40, p = 0.006), but there was no association with daytime sleepiness (Spearman's correlation, r = 0.04, p = 0.38) or depressive symptoms (Kruskal Wallis, KW statistic = 0.82, p = 0.66).

## Discussion

The present study investigated factors associated with chronic pain in a cohort of 614 male Southern African truck drivers. We took a holistic approach, and this is the first study to investigate general mental health, stress and sleep-related factors for associations with chronic pain in truck drivers. This was also the first study to investigate the association between working nights and pain. Working two or more nights a week and lower leisure-time physical activity was associated with having chronic pain. Results of this exploratory analysis showed that those driving two or more nights a week had worse sleep quality, greater daytime sleepiness and greater depressive symptoms. Sleep quality was also worse in those doing less leisure-time physical activity.

Pain and sleep have a reciprocal relationship with impaired sleep predicting pain, and pain impairing sleep [22,23]. Disrupted sleep and wake times may increase pain [51] and irregular bedtimes have associated with greater pain intensity in those with chronic pain [51]. Long-distance truck drivers have reported irregular working hours [19,52] that resulted in irregular sleep-wake times [19]. In the present study, only average sleep-and wake-times were recorded over the previous month, hence the regularity/irregularity of the participants' day-to-day sleep-wake schedules is unclear. We inferred, however, that individuals working two or more nights a week would have irregular sleep- and wake- times. Thus, we hypothesize that the mechanism through which working nights may have influenced pain is via circadian misalignment.

In the present study, about 10% of the participants had chronic pain. This prevalence is lower than the 27–81% reported in previous studies [17–20]. The wide range of pain prevalence may be due to several reasons: the older age of other cohorts [8,18,19], the inclusion of both acute and chronic pain, and the varying definitions of pain, such as "*any* ache pain or discomfort in the last year" to "pain often or all the time in the 12 months prior" [18,19]. Still, long-distance truck drivers are known to be more at-risk for chronic pain than short distance truckers [19]. For instance, a recent study performed among 123 truck drivers working for the same company in Canada reported that prevalence of musculoskeletal pain was twice as high among long-distance drivers compared to short-distance truckers [19]. Interestingly, the authors found that musculoskeletal pain among long-distance drivers was related to lifestyle (physical inactivity, smoking, reduced sleep) and psychosocial factors, while musculoskeletal pain among short-distance drivers was mainly related to physical risk factors (lifting and vibration).

Worldwide, including in South Africa, long-distance truck drivers are subject to challenging working conditions including long working hours, pressure to meet deadlines, insufficient rest, and exposure to violence such as high jacking of cargo [8,28,52–54]. In addition to increasing the risk of road accidents and chronic pain, working night shifts may promote stress and depressive symptoms [55]. Indeed, a meta-analysis found depression and night shift work were associated and that depression could be 40% more likely in those working night shifts [56]. In the present study, the prevalence of depressive symptoms was higher in the truck drivers than males in the general South African population (40% vs 24%) [57]. This finding suggests that assessment of mental health in truck drivers, and potential referral, could be a useful

addition to roadside health services. Moreover, there was a relationship between working two or more nights a week and greater depressive symptoms, and this result is in accordance with current literature [56]. In the same way as for chronic pain, we hypothesize that working nights may also have been associated with depression via chronic circadian misalignment [58].

Clinical evidence elucidating a mechanism between low mood and irregular sleep-wake schedules include data from 23 participants subjected to the forced desynchrony protocol, which teases apart endogenous circadian rhythms from the sleep-wake cycle [59]. A significant variation in the modulation of happiness was seen at different phases of the circadian rhythm: higher happiness coincided with core body temperature peak, and lowest happiness occurred at core body temperature trough, a time when normally-entrained individuals are asleep and therefore unlikely to report feeling less happy. Irregular sleep-wake schedules such as those seen in rotating shift workers may thus result in being awake at a time when mood is more depressed. Further, irregular sleep-wake schedules lead to exposing the circadian pacemaker to light at exquisitely sensitive parts of the phase response curve, which may lead to large phase shifts even with little light exposure [60,61]. This results in frequent phase shifts and loss of amplitude of endogenous rhythms in both rodents [62] and humans [63]. Flattening of endogenous rhythms is associated with poorer sleep quality, which may spill over even on days off work. Indeed, we found an association between working more than two night shifts per week and lower sleep quality. Similarly, rhythms of mood regulation may be flattened, which may make depression/mood dysregulation more likely. The likelihood of being more depressed when working nights, therefore, may be either a direct effect of dysregulation of mood through lower circadian amplitude and/ or an indirect effect elicited by the lower quality of sleep in individuals experiencing chronic circadian misalignment.

Despite relationships between pain, sleep and mood [49], and between stress, sleep, and mood [55], and general comorbidity between chronic pain and stress [64], we did not find an association between markers of stress (PTSD, occurrence of traumatic life events, and potential driving-related stressors like carrying dangerous cargo) and chronic pain. Stress, therefore, may not be part of the mechanism linking pain, sleep and mood in this cohort. The one non-work-related psychosocial factor that has previously been investigated in studies on pain in truck drivers, however, is time away from family. Extended time away from family was associated with having pain in Japanese drivers [65]. Anecdotally, Canadian drivers disliked spending large periods of time alone [19]. These data suggest that loneliness, which has associated with pain [66–68], may be worthy of investigation in future studies on pain in long-distance drivers including its impact on mood and sleep.

We found that lower leisure-time activity was associated with having chronic pain. Individuals with chronic pain had a median leisure-time physical activity score of 0 compared to a score of 17 but both groups still classify as being insufficiently active [69]. The truck drivers reported driving for a median of ten hours per day regardless of their chronic pain status. As our data are cross-sectional, we cannot report whether lower activity led to chronic pain or whether those with chronic pain were unable or unwilling to exercise. Physical activity may be associated with pain mitigating effects either directly through exercise-induced hypoalgesia [70] and/ or through its known circadian phase shifting effects [71,72]. Either way, physical activity is an integral component of pain management programs [73], so increasing physical activity, whether as a preventative or therapeutic strategy could be beneficial in populations of truck drivers for more than just their cardiovascular health [74]. Truck drivers with pain may, however, require professional support to ensure that activity is at an appropriate level/pace.

In an exploratory analysis we investigated whether leisure-time activity levels associated with sleep or depressive symptoms. Whilst we did not find an association with leisure-time activity and depressive symptoms or sleepiness, we did find that those who were less active had

worse sleep quality; a finding also reported in a meta-analysis of the effects of physical activity on sleep [75]. We have already implicated the role of sleep irregularity on the presence of pain and depressive symptoms but there is also evidence that sleep irregularity associates with greater functional impairment in those living with chronic pain [51]. The mechanisms influencing the relationships between physical activity, sleep, depression and pain certainly warrant further investigation.

The present study has provided data for the first time on the relationships between chronic pain, sleep, depression and physical activity in long-distance truck drivers. These data support interventions to improve physical and psychosocial health of long-distance truck drivers who are an important part of both the local and global economies. The drivers were recruited at truck stops where a health programme is already running. Health programmes dedicated to truck drivers exist in Sub-Saharan Africa too but most are dedicated to HIV prevention and diagnosis [26]. Wellness centres and exercise rooms have been proposed at trucker stops [76] and these data support such interventions. Health programmes could be expanded to assess mental health with drivers referred as necessary. The one difficulty in South Africa, however, is that most long-distance drivers are non-South Africans (whereas in other countries drivers are largely citizens), which has implications for retaining drivers in care, especially where chronic care is required. This issue can be overcome as illustrated by a Trucking Association having an electronic health passport system and a strong referral network for continuous access across major routes [77].

There were several limitations to the work. The protocol for recruitment included handing out leaflets at truck stops and addressing several truck drivers at a time about the study. We cannot exclude the possibility, therefore, of selection bias in our sample. We only recorded average sleep and wake times and so we do not know how interrupted drivers' sleep-wake schedules actually were. We would recommend the use of week-long sleep diaries in future to ascertain sleep irregularity. Additionally, there is poor agreement between subjective and objective measures of both sleep [78] and activity [79–81]. Objective measurement of both sleep and activity would strengthen future studies. Acceptability of objective measurement of sleep and has been demonstrated in British truck drivers and in clinical populations in South Africa [81,82]. We classified individuals as not having chronic pain if they had not had pain for at least three months. There were individuals, however, who had experienced pain for less than three months and who may have gone on to develop chronic pain. This issue may have reduced the likelihood of determining associations with pain. Nonetheless, working nights and activity were associated with chronic pain.

Chronic pain is more prevalent in the United States than diabetes, cardiovascular disease and cancer combined [16], and results in disability and reduced quality of life [12,13]. In this first study of factors associated with chronic pain in male long-distance truck drivers, working two or more nights a week and lower leisure-time physical activity were associated with having chronic pain. The findings here support the inter-relatedness of pain, sleep, mood and physical activity as working nights also associated with worse sleep and greater depressive symptoms, and lower leisure-time activity associated with worse sleep quality. Potential improvements to current health programmes for truck drivers include health screening that assesses pain, mental health, sleep and activity. The feasibility of exercise facilities at, or nearby, truck stops should also be explored. Since both sleep deprivation and pain could result in a multitude of negative consequences (including road accidents) stronger laws are recommended that govern working hours, number of hours on the road (possibly monitored through trackers in vehicles) and requirements on truck owners to ensure that their drivers have access to proper sleeping (and possibly food and exercise) facilities along their trucking routes. Truck driving contributes significantly the global economy and promoting health in truck drivers in these ways is

not only logical strategically but has the potential to reduce disability and improve quality of life in this at-risk occupational group [4].

## Acknowledgments

The authors would like to thank North Star Alliance for their assistance during participant recruitment and the truck drivers for their participation.

## Author Contributions

**Conceptualization:** Stella Iacovides, Johanna Roche, Karine Scheuermaier, W. D. Francois Venter, Alinda G. Vos, Samanta T. Lalla-Edward.

**Data curation:** Antonia L. Wadley, Johanna Roche, Karine Scheuermaier.

**Formal analysis:** Antonia L. Wadley, Johanna Roche, Karine Scheuermaier.

**Funding acquisition:** Alinda G. Vos, Samanta T. Lalla-Edward.

**Investigation:** W. D. Francois Venter, Alinda G. Vos, Samanta T. Lalla-Edward.

**Methodology:** Antonia L. Wadley, Stella Iacovides, Johanna Roche.

**Project administration:** Alinda G. Vos, Samanta T. Lalla-Edward.

**Resources:** W. D. Francois Venter.

**Visualization:** Stella Iacovides.

**Writing – original draft:** Antonia L. Wadley.

**Writing – review & editing:** Antonia L. Wadley, Stella Iacovides, Johanna Roche, Karine Scheuermaier, W. D. Francois Venter, Alinda G. Vos, Samanta T. Lalla-Edward.

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
