## [Decision Letter · Decision Letter 0]

21 Sep 2020

PONE-D-20-22799

Working nights and lower leisure-time physical activity associate with chronic pain in Southern African truck drivers: a cross-sectional study

PLOS ONE

Dear Dr. Wadley,

Thank you for submitting your manuscript to PLOS ONE. After careful consideration, we feel that it has merit but does not fully meet PLOS ONE’s publication criteria as it currently stands. Therefore, we invite you to submit a revised version of the manuscript that addresses the points raised during the review process.

Both reviewers noted that the introduction required additional rationale and lieterautre support and the discussion required additional explanation for the real-world application of the resutls. Finally, the potential confound mentioned by reviewer 1 must be addressed.  

We look forward to receiving your revised manuscript.

Kind regards,

Jennifer Teske

Academic Editor

PLOS ONE

Journal Requirements:

Reviewers' comments:

Reviewer's Responses to Questions

**Comments to the Author**

1. Is the manuscript technically sound, and do the data support the conclusions?

Reviewer #1: Yes

Reviewer #2: Yes

2. Has the statistical analysis been performed appropriately and rigorously? 

Reviewer #1: Yes

Reviewer #2: Yes

3. Have the authors made all data underlying the findings in their manuscript fully available?

Reviewer #1: Yes

Reviewer #2: Yes

4. Is the manuscript presented in an intelligible fashion and written in standard English?

Reviewer #1: Yes

Reviewer #2: Yes

5. Review Comments to the Author

Reviewer #1: I found this to be an interesting manuscript that certainly addresses a gap in the literature. Before being accepted for publication, there are several areas in my view that need to be addressed by the authors:

General Comments/Feedback:

-There are opportunities for improving language/grammar in the manuscript. For example, line 79 should read "associated with" rather than "associating with".

Introduction:

-Overall, the section seems a bit thin...there isn't a strong theoretical or practical (i.e., for prevention/intervention) basis/justification for exploring the relationships that are queried in this study

-There is a reference in your first paragraph for the "American Trucker Association" - did the authors mean the American Trucking Associations?

-There are seemingly direct quotes that should have adequate in-text citations to know which studies each came from.

Materials and Methods:

-More information regarding how participants were recruited, what day/time drivers were recruited, the percent of drivers that were approached that agreed to participate, etc. would be helpful.

-It isn't clear whether participants were offered incentives.

-If drivers did not fast, it may have possibly impacted their c-reactive protein levels (leading to an important confound). I am not certain whether this is the case, but this should be looked into by the authors and, if this indeed a possible confound, it should be acknowledged as a limitation.

-In the United States, there are many different types of truck driver routes (e.g., long-haul), and the type of route can have important implications for health and safety. Is this also the case in South Africa? If so, the types of drivers that participated and how these characteristics may have impacted the findings should be discussed in the manuscript.

Discussion:

-It would be helpful to readers outside of the South African context to provide further insight into the trucking industry in South Africa, and especially the key similarities and differences with other trucking industries globally.

-The authors should reflect on the findings of their work in reference to truck driver health and wellness prevention/intervention programs - what do these findings contribute to actually improving the quality of life of professional drivers?

Reviewer #2: This is a really interesting study of the association of chronic pain truck drivers with work factors and health behaviors/outcomes. The reviewer appreciates the opportunity to review this manuscript. There are several areas for improvement with the recommendations provided below by section of the manuscript.

Introduction:

- The authors need to provide much more background literature in relation to the working conditions of truck drivers.

- The reviewer suggests that the authors do not include their recent study in the introduction - move it and refer to it in the discussion section.

- The authors should not include the last paragraph of the introduction - it should be moved to the materials and methods section.

- Overall, the authors need to provide much more of a rationale for the importance of the study - much more literature support needs to be provided.

Materials and methods:

- The authors need to provide more details about the data collection process. How were the truck stops chosen? Why?

- More details need to be provided in the "Chronic pain" section. Was a valid and reliable scale used, etc.?

- More rationale needs to be provided for the "exploratory analyses", both in the introduction and analyses section. Why was it included in the present study?

Results:

- Overall, no major concerns regarding how the results are presented. Easy to follow!

Discussion:

- Authors use "We" too much - refer to it as "The present study" or something similar.

- The authors do a good job of relating their findings to previous studies; however, the authors need to do a better job and expand more on the real-world implications of the findings. The reviewer recommends that the authors provide much more in terms of the implications of the findings. What are potential viable solutions for addressing the issues among truck drivers in South Africa?

- The reviewer recommends a "Conclusions" section that gives a final "big picture" implications of the study and findings.

6. PLOS authors have the option to publish the peer review history of their article (what does this mean?). If published, this will include your full peer review and any attached files.

Reviewer #1: **Yes: **Michael Kenneth Lemke

Reviewer #2: No

---

## [Author Response · Author response to Decision Letter 0]

9 Oct 2020

Dear Prof Teske, 

Re: PONE-D-20-22799

Working nights and lower leisure-time physical activity associate with chronic pain in Southern African truck drivers: a cross-sectional study

Thank you very much for the time spent by you and the reviewers reviewing our manuscript. We are extremely grateful for the insights and comments, which we believe have really strengthened our manuscript. 

We have complied with all the reviewers’ comments and include our responses point by point in the response to reviewers document. Thank you for re-considering our manuscript for publication. 

Yours sincerely, 

Antonia Wadley (on behalf of all the authors)

---

## [Decision Letter · Decision Letter 1]

20 Nov 2020

Working nights and lower leisure-time physical activity associate with chronic pain in Southern African long-distance truck drivers: a cross-sectional study

PONE-D-20-22799R1

Dear Dr. Wadley,

We’re pleased to inform you that your manuscript has been judged scientifically suitable for publication and will be formally accepted for publication once it meets all outstanding technical requirements.

Kind regards,

Jennifer Teske

Academic Editor

PLOS ONE

Additional Editor Comments (optional):

Reviewers' comments:

Reviewer's Responses to Questions

**Comments to the Author**

1. If the authors have adequately addressed your comments raised in a previous round of review and you feel that this manuscript is now acceptable for publication, you may indicate that here to bypass the “Comments to the Author” section, enter your conflict of interest statement in the “Confidential to Editor” section, and submit your "Accept" recommendation.

Reviewer #2: All comments have been addressed

2. Is the manuscript technically sound, and do the data support the conclusions?

Reviewer #2: Yes

3. Has the statistical analysis been performed appropriately and rigorously? 

Reviewer #2: Yes

4. Have the authors made all data underlying the findings in their manuscript fully available?

Reviewer #2: Yes

5. Is the manuscript presented in an intelligible fashion and written in standard English?

Reviewer #2: Yes

6. Review Comments to the Author

Reviewer #2: As with the previous review, the authors have conducted a really good study. The reviewer commends the authors for their improved manuscript. The authors have done a good job of expanding in the introduction, providing further details in the methods section, and expanding the discussion/conclusion section. At this point, the authors have addressed the reviewer's concerns and feels that the manuscript is ready for publication. Good job, team!

7. PLOS authors have the option to publish the peer review history of their article (what does this mean?). If published, this will include your full peer review and any attached files.

Reviewer #2: No

---

## [Editor Report · Acceptance letter]

24 Nov 2020

PONE-D-20-22799R1 

Working nights and lower leisure-time physical activity associate with chronic pain in Southern African long-distance truck drivers: a cross-sectional study 

Dear Dr. Wadley:

I'm pleased to inform you that your manuscript has been deemed suitable for publication in PLOS ONE. Congratulations! Your manuscript is now with our production department. 

Kind regards, 

on behalf of

Dr. Jennifer Teske 

Academic Editor

PLOS ONE